# Evaluation of Palatal Bone Thickness at the Implantation Areas of Two Popular Bone-Anchored Distalizers—A Cone Beam Computed Tomography Retrospective Study

**DOI:** 10.3390/diagnostics13142421

**Published:** 2023-07-20

**Authors:** Marta Gibas-Stanek, Szczepan Żabicki, Michał Urzędowski, Małgorzata Pihut

**Affiliations:** Department of Prosthodontics and Orthodontics, Dental Institute, Faculty of Medicine, Jagiellonian University Medical College, Montelupich St. 4/108, 31-155 Krakow, Poland; szczepanzabicki@gmail.com (S.Ż.); michal.urzedowski@gmail.com (M.U.); malgorzata.pihut@uj.edu.pl (M.P.)

**Keywords:** bone screw, maxilla, orthodontics, X-ray computed tomography

## Abstract

Since class II malocclusion and lack of space within the dental arch due to early loss of deciduous molars is a common orthodontic problem in the Polish population, bone-anchored distalizers are becoming more and more popular. The aim of the present study was to evaluate palatal soft and hard tissue thickness using cone beam computed tomography (CBCT) at the area of micro-implant placement of two appliances for maxillary first molar distalization: Beneslider and TopJet distalizer. The study data were 100 consecutively selected CBCT images (53 of men and 47 of women). Measurements of bone and mucosa thickness were performed at six locations in the palate and tested according to their correlation with sex and age. The biggest bone and mucosa thickness were recorded in the insertion site of the TopJet miniscrew. Bone thickness in all points of paramedian insertion was significantly greater in males and the mean difference was approximately 1–1.8 mm. Age correlates significantly (*p* < 0.05) and positively (r > 0) with the thickness of the mucosa at all points: the older the patient, the thicker the mucosa at each measurement point. Anatomical diversity of the hard palate in the population involves the need to perform bone and mucosa thickness measurements before palatal micro-implant placement.

## 1. Introduction

Skeletal anchorage irreversibly changed the process of orthodontic treatment planning enabling tooth movements, previously considered impossible or at least difficult. It also minimized the need for patient cooperation in terms of wearing extraoral appliances or intermaxillary elastics commonly recommended as a method of anchorage reinforcement [1]. Molar distalization requires special attention in the context of maintaining anchorage as, regardless of the method used, in most cases, traditional intraoral distalizers lead to an undesirable proclination of the teeth from the anterior anchor unit [2,3,4,5,6,7,8]. Moreover, distalization is frequently associated with adverse effects in the vertical plane such as molar extrusion and an increase in the lower facial height [9,10,11].

Since class II malocclusion and lack of space within the dental arch due to early loss of deciduous molars is a common orthodontic problem in the Polish population, bone-anchored distalizers are becoming more and more popular [12,13]. Maximum anchorage and no need to rely on the patient’s cooperation make it a valuable alternative to traditional appliances [14]. The palatal location of mini-implants minimizes the risk of root injury [15]; however, the thickness of the bone should be known to provide the mechanical retention of the miniscrew as well as avoid nasal cavity and nasopalatine canal perforation. Moreover, the stability of palatal micro-implants inserted in the anterior region of the maxilla reaches 98% and exceeds miniscrews survival rate in other locations [1,16].

The aim of the present study was to evaluate palatal soft and hard tissue thickness using cone beam computed tomography (CBCT) at the area of micro-implant placement of two appliances for maxillary first molar distalization, that are successfully used in the Department of Prosthodontic and Orthodontics of Jagiellonian University in Cracow: Beneslider (PSM Medical Solutions, Tuttlingen, Germany) and TopJet distalizer (Tiger Dental, Bregenz, Austria) (Figure 1 and Figure 2).

Beneslider is a distalization appliance supported with two orthodontic mini-implants (7, 9, or 11 mm long) positioned medially or paramedially in the anterior palate [2,3]. The TopJet distalizer is a prefabricated appliance anchored by a single miniscrew (10, 12, or 14 mm long) in the anterior palate, laterally to the midpalatal suture [4]. Although cortical bone thickness determines the primary stability of micro-implants [5,9,10], it is essential to know total bone and soft tissue thickness to choose the appropriate implant length. In light of this fact and the lack of studies comparing implantation conditions for these two popular distalizers, the present study was carried out to provide clinically relevant information allowing to choose an adequate distalization system. Additionally, thus far, no studies have been conducted in the Polish population to assess palatal bone thickness. The hypothesis of the present study was that the palatal bone and mucosa thickness tends to increase from medial to lateral regions and decrease from anterior to posterior areas of the hard palate. Since both young and adult patients are seeking orthodontic treatment, another objective was to evaluate dependency between bone availability, age, and gender.

## 2. Materials and Methods

### 2.1. The Source of Data

The source of data for this cross-sectional retrospective research was cone-beam computed tomography scans of the maxilla of patients of University Dental Clinic in Krakow performed for any reason between 2018 and 2021. The study data were 100 consecutively selected CBCT images (53 of men and 47 of women) that met the inclusion criteria: age of the patient > 11 years, presence of upper first and second premolars, absence of pathologies in the anterior part of the maxilla (such as cysts, supernumerary teeth, impacted teeth), and absence of clefts and maxillofacial syndromes. Patients with a history of facial surgery or facial trauma and images with artifacts were excluded from the study. 

The CBCT scans were obtained using the OP 3D Pro (KaVo, Berlin, Germany). Average parameters: field of view 130 × 150 mm, average exposure time 8.5 s, average scanning time 39 s, and average voxel size 380 μm–5 mA. All the images were analyzed by a single trained and calibrated senior postgraduate trainee in orthodontics. Intra-examiner error was calculated by repetitive measurements within a 3-week interval on 10 randomly selected scans to determine the reliability. For the analysis of the CBCT images, a medical diagnostic monitor (RadiForce MX215, EIZO, Viena, Austria) and InViVo Dental Viewer (Anatomage, Santa Clara, CA, USA) were used.

### 2.2. Measurements

The measurements were performed in three planes of space: the horizontal plane passing through the right and left Orbitale and right and left Porion points;the frontal plane passing through the right and left Porion point, perpendicular to the horizontal plane; andthe sagittal plane passing through the Nasion point, perpendicular to the horizontal and frontal planes.

Measurements of bone and mucosa thickness were performed at six locations in the palate. In the first part of the study, tissue thickness for TopJet miniscrew was measured as presented in Figure 2a,b. The line passing through the middle of the oral opening of the nasopalatine canal and posterior nasal spine was designated as the midline of the palate (x-axis). The line passing through the midpoints of the crowns of the first premolars was determined as the p1-axis. The tissue thickness for TopJet microscrew was measured in coronal view at the M4 point (half-distance from the palatal cusp of the first premolar to the midline of the palate) [4] when the scan was oriented according to the p1-axis. The measurement was performed separately for the soft and hard tissues. Since previous studies presented no difference between the right and left side of the patient, only one side (right) was assessed for the purpose of this study [11,17].

In the second part of the study, tissue thickness for paramedian insertion of Benefit micro-implants was investigated as presented in Figure 2c,d. For the purpose of measurements, CBCT scans were positioned according to the p2 axis. Tissue thickness was assessed in the frontal plane at points B2.5 and B5 located 2.5 mm and 5 mm laterally to the x-axis and perpendicularly to the occlusal plane as recommended for paramedian insertion of Benefit miniscrews [12]. Soft and hard tissue thickness was measured.

In the last part of the study, tissue thickness for the median pattern of insertion of Benefit miniscrews was determined. According to the authors of the method, micro-implants should be placed along the x-axis 7–14 mm apart [12]. For the purpose of this measurement, the sagittal slice of CBCT was assessed when the scan was oriented according to p2 and x-axis. Mucosa thickness and palatal bone thickness were measured at the level of the p2 axis and 7 mm, and 14 mm distally, perpendicularly to the palate, as presented in Figure 2e,f.

The abovementioned measurements were tested according to their correlation with sex and age.

The study was approved by the bioethics committee of Jagiellonian University (number 1072.6120.132.2020).

### 2.3. Statistical Analysis

Data analysis was performed using R software version 4.2.1 (R: A language and environment for statistical computing. R Foundation for Statistical Computing, Vienna, Austria).

The normality of the distribution of the various groups was tested by means of a Shapiro–Wilk test. Analysis of quantitative variables was performed by calculating mean, standard deviation, median, and quartiles. Qualitative variables were analyzed by calculating the number and percentage of occurrences of each value. A Mann–Whitney test was used to compare quantitative variables between two groups. The relationship between quantitative variables was assessed with Spearman’s coefficient of correlation. A comparison of the values of quantitative variables in three repeated measurements (measurement points) was made using the Friedman test. After detecting statistically significant differences, a post-hoc analysis (Wilcoxon matched-pairs test with Bonferroni correction) was performed in order to identify statistically significantly different measurements. The significance level for all statistical tests was set to 0.05.

## 3. Results

### 3.1. Patient Characteristics

A total of 100 consecutively selected CBCT images of the maxilla that met the inclusion criteria were assessed in this study. The scans came from 47 women and 53 men with a mean age of 31.9 years for women and 31.7 years for men (Table 1). The age of the patients ranged from 11 to 76 years.

### 3.2. Tissue Thickness

Ten randomly selected scans were subjected to repeated measurements within a 3-week interval. The concordance of the measurements of quantitative variable assessment with intraclass correlation coefficient type 2 (according to the Shrout and Fleiss classification) indicated excellent accordance between the first and second measurements.

Values of the thickness of hard and soft tissues as well as general tissue depth are presented in Table 2, Figure 3 and Figure 4. The biggest paramedian tissue thickness as well as bone and mucosa thickness were recorded in the M4 insertion site (Table 2). Obtained M4 values were almost twice as high as the results of B2.5 thickness. Comparing insertion sites for Benefit miniscrews, general tissue thickness, and mucosa thickness was greater at the level of B5 point, but bone thickness was comparable with even slightly greater values in the case of B2.5 point.

In the case of median insertion sites, the biggest general tissue thickness was recorded at the level of point B (6.68 ± 1.67 mm) but the greatest bone thickness was observed at B14 point (5.01 ± 2.06 mm). Mucosa thickness was gradually decreasing from point B (1.91 ± 0.7 mm) to point B14 (1.39 ± 0.96 mm).

### 3.3. Correlation with Sex and Age

Considering the correlation with the sex of the patients, bone thickness in all points of paramedian insertion was significantly greater in males and the mean difference was approximately 1–1.8 mm (Table 3).

In the case of mucosa thickness, the correlation was not so clear, and a statistically significant difference was found only in the case of B7 and B14 points (Table 4).

Age correlates significantly (*p* < 0.05) and negatively (r < 0) with the bone thickness only at the B14 point, so the older the age, the thinner the bone at this point (Table 5, Figure 5). Age correlates significantly (*p* < 0.05) and positively (r > 0) with the thickness of the mucosa at all points: the older the patient, the thicker the mucosa at each measurement point (Table 6, Figure 6).

## 4. Discussion

The increasing popularity of bone-anchored palatal appliances is attributed to their effectiveness and lack of common side effects associated with traditional distalizers [5]. Although palatal micro-implants are considered to have a relatively low failure rate, their stability depends on bone thickness which should be greater than 5 mm [13,18].

Considering paramedian sites for micro-implant insertion, the greatest values of bone thickness were found at the M4 point (8.01 ± 3.53 mm). Significantly thinner bone levels were located at points B2.5 and B5 (4.57 ± 2.21 mm and 4.28 ± 2.36 mm, respectively). Amri et al., who measures bone thickness 3 mm and 6 mm laterally to the midsagittal line at the level of contact points of first and second premolars, obtained similar results (4.51 mm and 3.65 mm) and concluded that palatal bone thickness tends to decrease from medial to lateral region [14]. Nevertheless, measurements performed in the same study 9 mm laterally to the midsagittal suture seem to increase again, which also supports the results of our research. A similar tendency is noticeable in the results obtained by Suteerapongpun et al.: the thickest palatal bone in the premolars region was found 9 and 12 mm laterally to the midpalatal suture [17]. Taking into account mucosa thickness, values tend to increase significantly from medial (2.40 ± 1.23 mm at B2.5 point) to lateral area (5.25 ± 1.85 mm at M4 point). Consequently, greater general tissue thickness at the level of M4 point (13.26 ± 3.81 mm) is associated with the necessity of choosing longer micro-implants when compared to B2.5 and B5 regions (6.97 ± 2.65 mm and 7.68 ± 3.03 mm). 

To minimize the potential risk of incisor root damage, median location of palatal micro-implants along the midpalatal suture is also recommended [12]. However, this area of implantation should be avoided in children and adolescents as a site of maxillary bone growth. Moreover, incomplete mineralization of the suture may affect micro-implant stability and increase the probability of miniscrew loss [18]. Assessing bone thickness in the median area, we obtained values similar to paramedian insertion sites (from 4.44 mm in B7 point to 5.01 mm in B14 point). Quite different results are presented in a study from Taiwan, where median measurements were significantly greater than bone thickness 2 and 4 mm laterally to midpalatal suture [19]. Mean bone thickness in the suture ranged from 7.07 mm at the point 6 mm posterior to the incisive foramen to 9.04 mm at the point 22 mm posterior to the incisive foramen, which gives values greater than those obtained in our study. Marquezan et al. [20] found that bone thickness in the suture ranged from 7.58 mm at the level of first premolars to 5.13 mm at the level of first molars, which is clearly in contradiction with the study from Taiwan showing decreasing tendency from anterior to posterior. Chang et al. [21] noticed that palatal bone thickness obtained in Asian populations was thicker compared to studies conducted in the Caucasian population, which could be confirmed by the results of the present study. The mean total tissue thickness in the medial area of the palate ranged from 5.9 mm in B7 point to 6.68 mm in B point; consequently, the preferable length of Benefit miniscrew should be (similarly to the paramedian location) equal to 7 mm. 

There are theories that micro-implant resistance to lateral load depends on the ratio of insertion depth to total miniscrew length [20]. As a result, ideal mucosa thickness should be minimal to obtain a small extrabony segment of the micro-implant and to increase resistance to lateral forces [18]. According to the results of the present study and another study conducted in our department on a different group of patients [21], the thinnest mucosa is located at the palatal suture and the values decrease with an increasing distance from the incisive foramen and increases laterally the further from the palatal suture line. The thinnest mucosa was localized at the B14 point (1.39 mm) and the thickest at the M4 point (5.25 mm), which significantly affects the extrabony length of miniscrew and requires the use of suitably longer miniscrews in the lateral parts of the palate comparing to the central region.

Considering the abovementioned results, it can be summarized that the initial hypothesis of our research was confirmed only to some extent. The thickness of the palatal mucosa tends to increase with the growing distance from the midpalatal suture and decrease from anterior to posterior areas, but in the case of bone thickness, correlation is not so obvious. Indeed, the greatest values were recorded laterally, in the middle of the palate, but bone at the suture and 2.5 mm laterally was thicker than 5 mm from the midline. Conversely, in the sagittal dimension, bone depth in the midpalatal suture seemed to increase with the growing distance from p2 line.

We were able to find a correlation between the gender of patients and palatal depth in points M4, B2.5, and B5 (males had significantly greater bone thickness), which is in line with other studies [10,11]. This correlation was not observed in the case of median measurements. Although there are studies where no correlation between gender and palatal bone thickness has been found [19], Mani et al. proved that the micro-implant failure rate was significantly higher in the case of females [22].

Most of the analyzed studies found no correlation between age and palatal bone thickness, which partially confirms the results of our study indicating poor negative age correlation only in the case of point B14 [14,18,20,21,23]. Nevertheless, Holm et al. [11] reported thicker bone in the anterior and median regions of the palate in the 9–13-year-olds compared to older patients. This might be attributed to the younger age of the participants and incomplete root development of canines and premolars combined with different study methodologies (measurements were performed perpendicularly to the bone surface both in median and paramedian regions). According to the results of the present research, age correlates positively with the thickness of the mucosa. This is in line with other studies [24,25,26]. Along with Song’s opinion [24], greater thickness of the palatal mucosa in older patients might be associated with the increase of the fat content in this tissue. However, there are also reports, where no dependency between age and mucosa thickness was found [20,27,28].

In light of the general guidelines regarding recommended minimum bone thickness equal 5 mm [13], according to the results of the present study, this value was not reached in the case of 55% of analyzed patients in point B, 59% in B7, 51% in B14, 60% in B2.5, and 65% in B5. Even though the most preferable conditions were found in the M4 point, in 22% of cases, bone thickness was smaller than 5 mm. Interestingly, in 5 cases where bone thickness in the paramedian area (point M4) was insufficient, the thickness of the bone at the suture was even greater than average. Due to the above, we can assume that bone volume in one area cannot be predicted by the mean of measurement in another point. Considering all potential points for implantation selected for the purpose of this research, in 15 cases, bone thickness did not reach a minimum of 5 mm.

The results of our study provide information regarding average bone thickness in the area of application of popular bone-borne distalizers; nevertheless, it should be interpreted in the light of certain limitations. One of them is a correlation between palatal bone thickness and skeletal relation in the sagittal and vertical plane: palatal bone thickness seems to be thinner in class III and high-angle patients compared to class I and normal and low-angle cases [17,29,30]. Another weakness concerns the age of the patients: both young and adult participants were included. Although numerous studies as well as the present study found no major correlation with age, it would be sensible to perform a comparison between patients with complete and incomplete facial growth. The third limitation is a relatively small sample size. 

## 5. Conclusions

The anatomy of the hard palate presents a considerable variety of shapes. In the studied group, the most preferable conditions for micro-implants were found at the point of insertion of the TopJet miniscrew (8.01 mm) and the thinnest bone was localized 5 mm laterally to the midpalatal suture (4.28 mm). In the case of 15 patients, bone depth did not reach a minimum of 5 mm at any of the analyzed points. Greater values of bone thickness in the paramedian areas were observed in the case of males. While no clear correlation was found between age and bone thickness, the thickness of the mucosa tends to increase with age. Although CBCT use in dentistry is sometimes controversial due to the relatively large radiation dose, anatomical diversity of the hard palate in the population involves the need to perform bone and mucosa thickness measurements before palatal micro-implant placement to select the appropriate size of the miniscrew and suitable area of the hard palate.

## Figures and Tables

**Figure 1 diagnostics-13-02421-f001:**
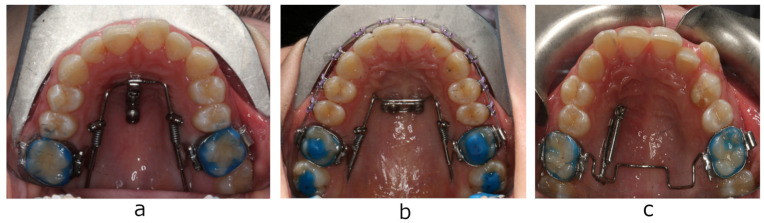
Bone-anchored distalizers: (**a**) Beneslider distalizer with mini-implants positioned medially; (**b**) Beneslider distalizer with mini-implants positioned paramedially; (**c**) TopJet distalizer.

**Figure 2 diagnostics-13-02421-f002:**
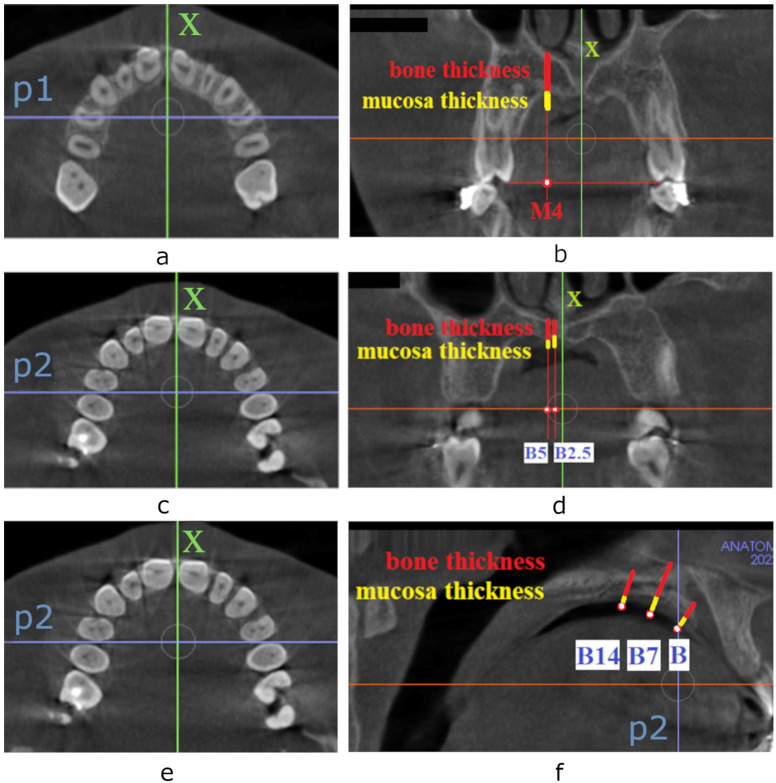
CBCT screenshots of the horizontal and frontal plane with the reference lines and points constructed for tissue thickness measurement. Horizontal (**a**) and frontal (**b**) view of the maxilla with reference lines and points constructed for tissue thickness assessment in the case of insertion of TopJet miniscrew: x—the midline of the palate, p1—the line passing through the midpoints of crowns of first premolars, M4 point—half-distance from the palatal cusp of the first premolar to the midline of the palate; CBCT screenshots of the horizontal (**c**) and frontal (**d**) plane with the reference lines and points constructed for tissue thickness assessment for paramedian insertion of Benefit micro-implants: x—the midline of the palate, p2—the line passing between first and second premolar, B2.5 point—a point located 2.5 mm laterally to the x-axis, B5—a point located 5 mm laterally to the x-axis; CBCT screenshots of the horizontal (**e**) and sagittal (**f**) plane with the reference lines and points constructed for tissue thickness assessment for median insertion of Benefit micro-implants: x—the midline of the palate, p2—the line passing between first and second premolar, B point—a point located at the level of p2 line, B7 point—a point located 7 mm distally to p2 line, B14—a point located 14 mm distally to p2 line.

**Figure 3 diagnostics-13-02421-f003:**
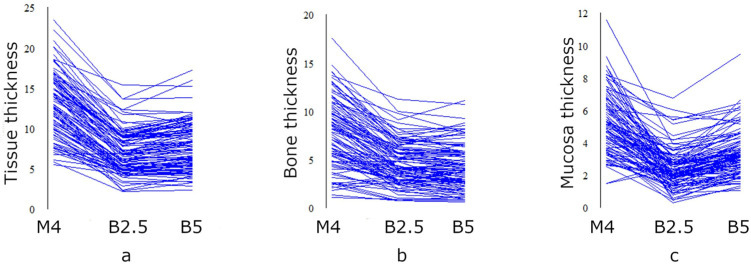
A parallel coordinate plot presenting general tissue thickness (**a**), bone thickness (**b**), and mucosa thickness (**c**) in points of paramedian insertion of TopJet miniscrew, Benefit miniscrew positioned 2.5 mm laterally, and Benefit miniscrew positioned 5 mm laterally.

**Figure 4 diagnostics-13-02421-f004:**
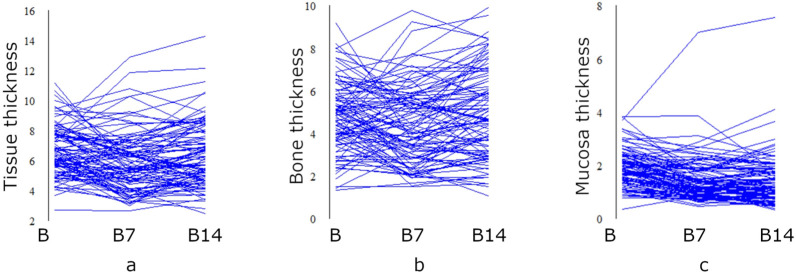
A parallel coordinate plot presenting general tissue thickness (**a**), bone thickness (**b**), and mucosa thickness (**c**) in points of median insertion of Benefit minisrew positioned in point B, 7 mm distally, and 14 mm distally.

**Figure 5 diagnostics-13-02421-f005:**
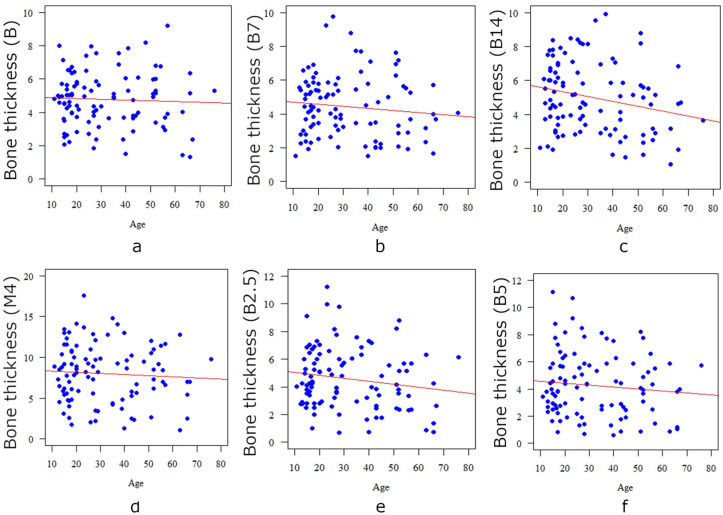
Correlation between the age of the patient and the bone thickness in the case of point B (**a**), point B7 (**b**), point B14 (**c**), point M4 (**d**), point B2.5 (**e**), and point B5 (**f**).

**Figure 6 diagnostics-13-02421-f006:**
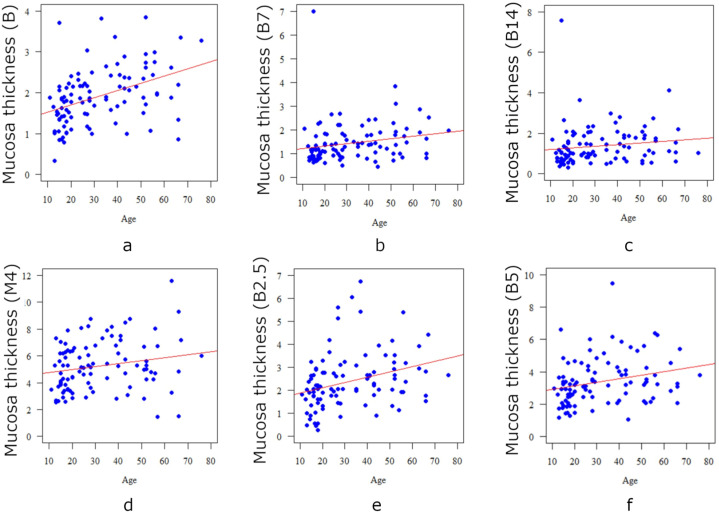
Correlation between the age of the patient and the mucosa thickness in the case of point B (**a**), point B7 (**b**), point B14 (**c**), point M4 (**d**), point B2.5 (**e**), and point B5 (**f**).

**Table 1 diagnostics-13-02421-t001:** Patients’ characteristics.

Sex	*N*	Age							*p*
		Mean	SD	Median	Min	Max	Q1	Q3	
Female	47	31.93	16.63	27	13	67	16.25	43.00	*p* = 0.898
Male	53	31.75	16.73	26	11	76	18.00	45.00	
Total	100	31.83	16.60	27	11	76	17.00	43.25	

**Table 2 diagnostics-13-02421-t002:** Comparison of tissue thickness for paramedian insertion of TopJet screw (Point M4) and Benefit System micro-implants (Points B2.5 and B5), and median insertion of Benefit System micro-implants located at the level of p2 line (Point B) and 7 mm and 14 mm posteriorly (Point B7 and B14).

Parameter	Measurement	*N*	Mean	SD	Median	Min	Max	Q1	Q3	*p*
Tissue thickness	Point M4	100	13.26	3.81	13.20	5.45	23.44	10.61	15.88	*p* < 0.001 * M4 > B5 > B2.5
	Point B2.5	100	6.97	2.65	6.68	2.15	15.41	4.87	8.93	
	Point B5	100	7.68	3.03	7.36	2.31	17.20	5.22	9.82	
	Point B	100	6.68	1.67	6.57	2.67	11.18	5.62	7.84	*p* < 0.001 * B, B14 > B7
	Point B7	100	5.90	2.00	5.56	2.66	12.91	4.56	6.93	
	Point B14	100	6.40	2.11	6.12	2.46	14.30	4.70	7.83	
Bone thickness	Point M4	100	8.01	3.53	8.20	1.05	17.57	5.53	10.57	*p* < 0.001 * M4 > B2.5 > B5
	Point B2.5	100	4.57	2.21	4.24	0.71	11.23	2.80	6.02	
	Point B5	100	4.28	2.36	3.96	0.60	11.14	2.51	5.76	
	Point B	100	4.77	1.64	4.88	1.33	9.19	3.69	5.75	*p* = 0.017 * B14 > B7
	Point B7	100	4.44	1.83	4.24	1.50	9.77	2.99	5.68	
	Point B14	100	5.01	2.06	4.78	1.07	9.90	3.17	6.50	
Mucosa thickness	Point M4	100	5.25	1.85	5.15	1.48	11.57	3.83	6.46	*p* < 0.001 * M4 > B5 > B2.5
	Point B2.5	100	2.40	1.23	2.19	0.27	6.74	1.62	2.91	
	Point B5	100	3.40	1.41	3.21	1.05	9.47	2.46	4.10	
	Point B	100	1.91	0.70	1.87	0.33	3.84	1.43	2.32	*p* < 0.001 * B > B7, B14
	Point B7	100	1.45	0.84	1.25	0.46	7.00	0.92	1.79	
	Point B14	100	1.39	0.96	1.11	0.31	7.57	0.78	1.77	

*p*—Friedman test + post-hoc analysis (paired Wilcoxon tests with Bonferroni correction). * statistically significant (*p* < 0.05)

**Table 3 diagnostics-13-02421-t003:** Palatal bone thickness and its correlation with the sex of the patients.

Parameter	Sex	*N*	Mean	SD	Median	Min	Max	Q1	Q3	*p*
Bone thickness (Point B)	Female	47	4.48	1.64	4.56	1.43	9.19	3.52	5.43	*p* = 0.071
	Male	53	5.02	1.62	5.10	1.33	8.21	3.98	6.05	
Bone thickness (Point B7)	Female	47	4.24	1.76	3.95	1.50	9.77	2.91	5.44	*p* = 0.299
	Male	53	4.62	1.88	4.97	1.52	9.24	3.23	5.78	
Bone thickness (Point B14)	Female	47	5.07	1.87	4.63	1.62	8.79	3.81	6.24	*p* = 0.735
	Male	53	4.95	2.24	5.09	1.07	9.90	2.93	6.59	
Bone thickness (Point M4)	Female	47	7.02	3.19	6.60	1.33	13.48	4.64	9.23	*p* = 0.009 *
	Male	53	8.88	3.62	9.17	1.05	17.57	6.73	11.43	
Bone thickness (Point B2.5)	Female	47	3.96	1.88	3.58	0.71	9.09	2.65	5.12	*p* = 0.008 *
	Male	53	5.11	2.35	5.22	0.73	11.23	3.52	6.73	
Bone thickness (Point B5)	Female	47	3.62	2.21	2.94	0.60	11.14	2.11	4.70	*p* = 0.006 *
	Male	53	4.86	2.36	4.59	0.85	10.68	3.10	6.44	

*p*—Mann–Whitney test, SD—standard deviation, Q1—lower quartile, Q3—upper quartile. * statistically significant (*p* < 0.05)

**Table 4 diagnostics-13-02421-t004:** Palatal soft tissue thickness and its correlation with the sex of the patients.

Parameter	Sex	*N*	Mean	SD	Median	Min	Max	Q1	Q3	*p*
Mucosa thickness (Point B)	Female	47	1.85	0.73	1.78	0.84	3.71	1.29	2.30	*p* = 0.229
	Male	53	1.97	0.68	1.95	0.33	3.84	1.58	2.31	
Mucosa thickness (Point B7)	Female	47	1.37	1.01	1.11	0.46	7.00	0.84	1.69	*p* = 0.03 *
	Male	53	1.53	0.67	1.37	0.73	3.85	1.04	1.84	
Mucosa thickness (Point B14)	Female	47	1.28	1.14	0.99	0.36	7.57	0.62	1.54	*p* = 0.022 *
	Male	53	1.48	0.76	1.35	0.31	4.11	0.91	1.87	
Mucosa thickness (Point M4)	Female	47	4.84	1.64	4.71	1.48	8.23	3.39	6.12	*p* = 0.078
	Male	53	5.63	1.96	5.27	2.52	11.57	4.26	6.90	
Mucosa thickness (Point B2.5)	Female	47	2.28	1.03	2.16	0.55	5.60	1.70	2.72	*p* = 0.621
	Male	53	2.50	1.38	2.23	0.27	6.74	1.62	3.22	
Mucosa thickness (Point B5)	Female	47	3.41	1.33	3.24	1.05	6.61	2.42	4.19	*p* = 0.676
	Male	53	3.39	1.49	3.16	1.19	9.47	2.46	4.04	

*p*—Mann–Whitney test, SD—standard deviation, Q1—lower quartile, Q3—upper quartile. * statistically significant (*p* < 0.05)

**Table 5 diagnostics-13-02421-t005:** Correlation between the bone thickness and the age of the patient.

Parameter	Age
	Spearman’s Correlation Coefficient
Bone thickness (Point B)	r = −0.029, *p* = 0.771
Bone thickness (Point B7)	r = −0.092, *p* = 0.365
Bone thickness (Point B14)	r = −0.216, *p* = 0.031 *
Bone thickness (Point M4)	r = −0.033, *p* = 0.743
Bone thickness (Point B2.5)	r = −0.131, *p* = 0.193
Bone thickness (Point B5)	r = −0.053, *p* = 0.602

* statistically significant (*p* < 0.05).

**Table 6 diagnostics-13-02421-t006:** Correlation between the mucosa thickness and the age of the patient.

Parameter	Age
	Spearman’s Correlation Coefficient
Mucosa thickness (Point B)	r = 0.461, *p* < 0.001 *
Mucosa thickness (Point B7)	r = 0.355, *p* < 0.001 *
Mucosa thickness (Point B14)	r = 0.307, *p* = 0.002 *
Mucosa thickness (Point M4)	r = 0.215, *p* = 0.031 *
Mucosa thickness (Point B2.5)	r = 0.374, *p* < 0.001 *
Mucosa thickness (Point B5)	r = 0.305, *p* = 0.002 *

* statistically significant (*p* < 0.05).

## Data Availability

The datasets generated and/or analyzed during the current study are available from the corresponding author upon reasonable request.

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
