# Peer review of "Evaluation of Palatal Bone Thickness at the Implantation Areas of Two Popular Bone-Anchored Distalizers—A Cone Beam Computed Tomography Retrospective Study"

_diagnostics, 2023, doi:10.3390/diagnostics13142421_

Round 1

Reviewer 1 Report

This research has a high potential to be published online in the Journal of Diagnostics. I have the following recommendations and suggestions

1. The intro part of the manuscript seems to be very short. It is recommended to add research questions and background of the research to enhance the interest of the readers

2. For the readers' clarity, adding more details in the captions of the figures is suggested. Furthermore, captions are usually given below the figures and for tables, usually the captions are provided above them.

3. The text in Figure 3 is very small, it is hard to read it. would be good if the text is a bit enlarged

4. Same is the case for the rest of the figures.. please enlarge the text in them

5. It is also suggested to enlarge the text of x and y axis in all figures

6.  Furthermore, it is also suggested to combine the two paragraphs in the conclusion section.. this is because usually, the conclusion is one paragraph..

Minor English language correction are to be done

Author Response

Dear Reviewer,

Thank you for giving me the opportunity to submit a revised draft of my manuscript titled “Evaluation of palatal bone thickness at the implantation areas of two popular bone-anchored distalizers- a cone beam computed tomography retrospective study” to the Journal of Diagnostics. I appreciate the time and effort that you have dedicated to providing your valuable comments on my paper. I have been able to incorporate changes to reflect most of the suggestions. I have highlighted the changes within the manuscript and here is a point-by-point response to your comments and concerns.

Comment 1. The intro part of the manuscript seems to be very short. It is recommended to add research questions and background of the research to enhance the interest of the readers

Response: Thank you for this suggestion. I have added an additional part and 5 new citations referring to the latest publications on the topic of molars distalization. The incorporated changes can be found on page 2, line 60.

 Comment 2. For the readers' clarity, adding more details in the captions of the figures is suggested. Furthermore, captions are usually given below the figures and for tables, usually the captions are provided above them.

Response: Thank you for pointing this out. I have corrected the location of the captions. I have also added more details in the captions of the figures.

Comment 3. The text in Figure 3 is very small, it is hard to read it. would be good if the text is a bit enlarged

Comment 4. Same is the case for the rest of the figures.. please enlarge the text in them

Comment 5. It is also suggested to enlarge the text of x and y axis in all figures

Response: Thank you for your comment on that. I agree that the text in the figures might have been hard to read and I have changed the size of the letter in all figures. Also, one of the Reviewers suggested reducing the number of figures and combining figures 1 and 2, and 3,4,5 in one picture. Amended figures can be found on pages 2 and 3.

Comment 6.  Furthermore, it is also suggested to combine the two paragraphs in the conclusion section.. this is because usually, the conclusion is one paragraph.

Response: Thank you for your suggestion. I have combined the two paragraphs and also rearranged the conclusion section according to the comment of another Reviewer. Incorporated changes can be found on page 13.

I look forward to hearing from you in due time regarding our submission and to respond to any further questions and comments you may have.

Sincerely

Marta Gibas-Stanek

Reviewer 2 Report

The research work on the evaluation of palatal soft and hard tissue thickness using cone beam computed tomography at the area of micro-implant placement. The topics is interesting and important for engineering application. There are some comments as follows:

1. The literature reviewing in Introduction should be elaborated, especially for the recent publications in the past three years.

2. The quality of Figure 6 and Figure 7 should be improved. Is it three sub-figures?

3. Each sub-figure in Figure 8 and 9 should be marked with the different states.

4. The discussion in section 4, the scientific merits of the obtained results should be highlighted.

5. The conclusions suggest to be re-writted  point-to point. And, some key data results should be presented.

Author Response

Dear Reviewer,

Thank you for giving me the opportunity to submit a revised draft of my manuscript titled “Evaluation of palatal bone thickness at the implantation areas of two popular bone-anchored distalizers- a cone beam computed tomography retrospective study” to the Journal of Diagnostics. I appreciate the time and effort that you have dedicated to providing your valuable comments on my paper. I have been able to incorporate changes to reflect most of the suggestions. I have highlighted the changes within the manuscript and here is a point-by-point response to your comments and concerns.

Comment 1. The literature reviewing in Introduction should be elaborated, especially for the recent publications in the past three years.

Response: Thank you for this suggestion. I have added an additional part and 5 new citations (numbers 6,7,8,15,16) referring to the latest publications on the topic of molars distalization. The incorporated changes can be found on page 2, line 60.

Comment 2. The quality of Figure 6 and Figure 7 should be improved. Is it three sub-figures?

Response: Thank you for your comment on that. I agree that the text in these figures might have been hard to read and I have changed them. There are sub-figures in each picture, so I put in letter designation (a, b and c). Also, one of the Reviewers suggested reducing the number of figures and combining figures 1 and 2, and 3,4,5 in one picture, so Figure 6 and Figure 7 changed the numbers to Figure 3 and Figure 4. They can be found on page 7 of the manuscript.

Comment 3. Each sub-figure in Figure 8 and 9 should be marked with the different states.

Response: Thank you for this suggestion. I put in letter designation and also improved the quality of the text in these figures. Figure 8 and 9 changed the numbers to Figure 5 and 6 and can be found on pages 10 and 11.

Comment 4. The discussion in section 4, the scientific merits of the obtained results should be highlighted.

Response: According to your suggestion I have expanded on that part in the revised manuscript (page 12, line 295).

I enclose added paragraph:

Considering the abovementioned results it can be summarized, that the initial hypothesis of our research was confirmed only to some extent. The thickness of the palatal mucosa tends to increase with the growing distance from the midpalatal suture and decrease from anterior to posterior areas, but in the case of bone thickness correlation is not so obvious. Indeed, the greatest values were recorded laterally, in the middle of the palate, but bone at the suture and 2,5 mm laterally was thicker than 5 mm from the midline. Conversely, in the sagittal dimension bone depth in the midpalatal suture seem to increase with the growing distance from p2 line.

Comment 5. The conclusions suggest to be re-writted  point-to point. And, some key data results should be presented.

Response: I have rearranged the conclusions section, however, I am not sure if I understood your suggestion correctly. According to the suggestions of another Reviewer, I have also combined the two paragraphs into one. I enclose the amended section:

The anatomy of the hard palate presents a considerable variety of shapes. In the studied group the most preferable conditions for micro-implants were found at the point of insertion of the TopJet miniscrew (8,01 mm) and the thinnest bone was localized 5 mm laterally to the midpalatal suture (4,28 mm). In the case of 15 patients bone depth did not reach a minimum of 5 mm at any of the analyzed points. Greater values of bone thickness in the paramedian areas were observed in the case of males. While no clear correlation was found between age and bone thickness, the thickness of the mucosa tends to increase with age. Although CBCT use in dentistry is sometimes controversial due to the relatively large radiation dose, anatomical diversity of the hard palate in the population involves the need to perform bone and mucosa thickness measurements before palatal micro-implant placement to select the appropriate size of the miniscrew and suitable area of the hard palate.

Please let me know If you decide that further corrections are necessary in that section.

I look forward to hearing from you in due time regarding our submission and to respond to any further questions and comments you may have.

Sincerely

Marta Gibas-Stanek

Reviewer 3 Report

Dear authors,

We have read with interest your manuscript untitled « Evaluation of palatal bone thickness at the implantation areas of two popular bone-anchored distalizers- a cone beam computed tomography retrospective study ».

The amount of clinical data is very interesting.

The main objective of the study is clear : « The aim of the present study was to evaluate palatal soft and hard tissue thickness using cone beam computed tomography »

We have some remarks, before accepting the manuscript for publication.

The introduction section provides convicing justification for the 6 investigated points on CBCT (M4 / B2.5 / B5 / B / B7 / B14)

However, when it comes to results, the emphasized distinctions between the 2 devices bring some confusion, and render difficult to interprate the results.

Why not grouping Table 2 and Table 3 in one large table, summarizing all the results of the 6 investigated points.

The multiplications of Figures are also not helping.

We recommend to group in 1 single figure the Figures 1 – 2, and, in the same concept, Figures 3 – 4 – 5 should be grouped in 1 figure.

Therefore, we recommend your manuscript to undergo minor revision before beeing suitable for publication.

Yours faithfully,

Minor editing of English language required

Author Response

Dear Reviewer,

Thank you for giving me the opportunity to submit a revised draft of my manuscript titled “Evaluation of palatal bone thickness at the implantation areas of two popular bone-anchored distalizers- a cone beam computed tomography retrospective study” to the Journal of Diagnostics. I appreciate the time and effort that you have dedicated to providing your valuable comments on my paper. I have been able to incorporate changes to reflect most of the suggestions. I have highlighted the changes within the manuscript and here is a point-by-point response to your comments and concerns.

Comment 1. Why not grouping Table 2 and Table 3 in one large table, summarizing all the results of the 6 investigated points.

Response: Thank you for this suggestion. I have combined Table 2 and Table 3 into one table and I agree, that it improved the ease of interpretation of the results. The modified Table 2 can be found in the revised manuscript on page 5.

Comment 2. We recommend to group in 1 single figure the Figures 1 – 2, and, in the same concept, Figures 3 – 4 – 5 should be grouped in 1 figure.

Response: According to your suggestion I have combined Figures 1-2 and Figures 3-4-5 in single figures. Following the instructions of another Reviewer I also enlarged the text in figures to make it more clear for the readers. Rearranged figures can be found in the revised manuscript on pages 2 and 3.

I look forward to hearing from you in due time regarding our submission and to respond to any further questions and comments you may have.

Sincerely

Marta Gibas-Stanek

Round 2

Reviewer 1 Report

The authors have made necessary amendmets.. the manuscript can be published online.